# A Comparison of Different Matrices for the Laboratory Diagnosis of the Epizootic American Foulbrood of Honey Bees

**DOI:** 10.3390/vetsci10020103

**Published:** 2023-02-01

**Authors:** Julia Ebeling, Antonia Reinecke, Niklas Sibum, Anne Fünfhaus, Pia Aumeier, Christoph Otten, Elke Genersch

**Affiliations:** 1Department of Molecular Microbiology and Bee Diseases, Institute for Bee Research, 16540 Hohen Neuendorf, Germany; 2Department for Biology and Biotechnology, Behavioural Biology and Biology Education, Ruhr-Universität Bochum, 44780 Bochum, Germany; 3Fachzentrum Bienen und Imkerei, Dienstleistungszentrum Ländlicher Raum Westerwald-Osteifel, 56727 Mayen, Germany; 4Institute of Microbiology and Epizootics, Department of Veterinary Medicine, Freie Universität Berlin, 14163 Berlin, Germany

**Keywords:** American Foulbrood, *Paenibacillus larvae*, diagnostic, brood comb honey, adult bees, hive debris

## Abstract

**Simple Summary:**

American Foulbrood is a bacterial disease of honey bee larvae caused by the spore-forming bacterium *Paenibacillus larvae*. It is a notifiable epizootic in most countries because it is highly contagious and lethal to affected colonies. For clinically diseased colonies, authorities often consider burning the only sustainable control measure, while infected but not yet diseased bee colonies can be cured by beekeeping measures. It is therefore of the utmost importance to identify such colonies. For the detection of *P. larvae* spores in infected colonies prior to the onset of clinical symptoms, different hive matrices can be used: brood comb honey, adult bees, seasonal hive debris, and winter hive debris. We evaluated the suitability of these matrices as sample materials for early *P. larvae* spore detection and demonstrate considerable differences in the growth of concomitant bacteria interfering with *P. larvae* detection and in the sensitivity and limit of detection of *P. larvae* spores. We conclude that brood comb honey and adult bees are equally well-suited as sample materials for the early detection of *P. larvae* spores, while hive debris samples should only be used when it is not possible to collect honey or adult bee samples from brood combs.

**Abstract:**

American Foulbrood (AFB) of honey bees caused by the spore-forming bacterium *Paenibacillus larvae* is a notifiable epizootic in most countries. Authorities often consider a rigorous eradication policy the only sustainable control measure. However, early diagnosis of infected but not yet diseased colonies opens up the possibility of ridding these colonies of *P. larvae* spores by the shook swarm method, thus preventing colony destruction by AFB or official control orders. Therefore, surveillance of bee colonies for *P. larvae* infection followed by appropriate sanitary measures is a very important intervention to control AFB. For the detection of *P. larvae* spores in infected colonies, samples of brood comb honey, adult bees, or hive debris are commonly used. We here present our results from a comparative study on the suitability of these matrices in reliably and correctly detecting *P. larvae* spores contained in these matrices. Based on the sensitivity and limit of detection of *P. larvae* spores in samples from hive debris, adult bees, and brood comb honey, we conclude that the latter two are equally well-suited for AFB surveillance programs. Hive debris samples should only be used when it is not possible to collect honey or adult bee samples from brood combs.

## 1. Introduction

The Western honey bee *Apis mellifera* is an indispensable insect pollinator for many agricultural fruit and crop plants, as well as wild flowers, and is therefore essential for human food security and biodiversity conservation [1,2,3,4]. Therefore, diseases affecting honey bees are of concern not only to insect specialists, but also to veterinarians and to humans in general [5,6].

American Foulbrood (AFB) is a lethal bacterial brood disease of the Western honey bee caused by the Gram-positive, spore-forming bacterium *Paenibacillus larvae* (*P. larvae*) [7]. The disease is not only fatal to infected honey bee larvae, but can also lead to the collapse of the entire colony, when the loss of brood can no longer be substituted by the queen through increased egg-laying activity. As a result, the number of adult bees in the colony continuously decreases and eventually becomes insufficient to sustain the colony. In addition to its virulence, *P. larvae* is also highly contagious, and the infectious spores are easily spread within a colony, within an apiary, and also between neighboring apiaries [8,9,10]. The spores of *P. larvae* are extremely persistent with respect to environmental conditions [11], making eradication of the spores and control of the disease difficult. For these reasons, AFB is classified as a notifiable epizootic in many countries, and in the event of an outbreak authorities often consider a rigorous eradication policy the only sustainable control measure.

An outbreak of AFB is defined by (i) the presence of specific clinical symptoms at colony level (ropy masses in capped brood cells and a so-called foulbrood scale tightly adhering to lower cell rims) and (ii) the detection of *P. larvae* in samples taken from the suspected colony. The sole detection of *P. larvae* in a not yet clinically diseased colony should be classified as a *P. larvae* infection and not an outbreak of AFB disease. Such infected but clinically unobtrusive colonies can be sanitized and cleared from *P. larvae* spores by preparing shook swarms or artificial swarms [9,12], a method that is not effective in clinically diseased colonies [9]. Therefore, surveillance of bee colonies for the presence of *P. larvae* spores, and hence *P. larvae* infection, followed by appropriate sanitary measures, is a very important non-pharmaceutical intervention to reduce the number of outbreaks and prevent further spread of AFB [13].

For the detection of *P. larvae* spores in infected colonies in the course of AFB surveillance programs, samples of different matrices, such as honey, adult bees, or hive debris, are collected from the colony in question and cultivated in the lab on appropriate culture media [14], followed by the confirmation of suspicious colonies as *P. larvae* colonies by, e.g., species-specific PCR protocols [7,15]. An important issue when culturing *P. larvae* from spores is that, although *P. larvae* germinates within hours in the larval gut [16], germination of *P. larvae* spores on culture plates takes approximately three days [14]. Therefore, other bacteria present in the sample (hereafter named concomitant bacteria), which germinate and grow faster, may interfere with the detection of *P. larvae* simply by reducing the space and nutrients available for *P. larvae* germination and growth. Thus, the suitability of a particular matrix for the detection of *P. larvae* spores depends not only on the amount of *P. larvae* spores expected to be present in that matrix, but also on the number and type of concomitant bacteria normally present in samples from that matrix.

Honey, or more precisely the honey arch surrounding the brood nest (so-called brood comb honey), is well established as a sample material in AFB monitoring programs, e.g., in Germany [17,18], as this honey perfectly reflects the “spore situation” in the brood nest. Spores of *P. larvae*, if contained in this honey, will contaminate the mouthparts of nurse bees, resulting in the infection of susceptible first-instar larvae, even if they are fed only glandular secretions at this stage of larval development. Similarly, nurse bees which encounter the first foulbrood scale and attempt to remove it become contaminated with *P. larvae* spores and carry them into the brood comb honey. Therefore, analysis of brood comb honey for *P. larvae* spores is considered a very sensitive method and even suitable for the early detection of infected but not yet diseased colonies.

Adult bees are commonly used as sample materials for AFB analysis in Sweden, but also in New Zealand and Switzerland [19,20,21,22,23]. Although adult bees are not susceptible to *P. larvae* infection, they become contaminated by *P. larvae* spores in infected and diseased colonies [23,24] when they clean cells from foulbrood scales and cannibalize diseased or dead young larvae: such cleaning activities are part of the social immune response against the disease [25] because spores ingested orally will be defecated outside the hive and thus no longer contribute to disease transmission within the colony. However, spores on the mouth parts and body surfaces of adult bees will contaminate the larval food and brood cells and are thus the main route of disease transmission within the colony. *P. larvae*-spore-contaminated adult bees can carry a large number of *P. larvae* spores, and larval mortality was shown to be closely related to the spore load of adult bees from the brood nest [26]. Therefore, adult bees are considered a highly suitable matrix for the detection of *P. larvae* in infected honey bee colonies.

Recently, it has been reported that winter hive debris can also be used for the detection of *P. larvae*-infected colonies, especially in spring, when cold weather conditions hamper the visual inspection of colonies [27,28]. Hive debris can be collected from the bottom board and represents the waste produced in the bee colony (remnants of cell cappings, wax, pollen, dead *Varroa destructor* mites, ants, etc.). The advantage of hive debris as an analysis matrix is that it can be sampled without opening the hive and disturbing the colony. A clear disadvantage is that it is waste and hence can be expected to contain many environmental and saprophytic bacterial and fungal species, which could interfere with the detection of *P. larvae*.

If *P. larvae* spores are detected in a honey bee colony based on the analysis of the above-mentioned hive matrices, the colony is considered infected; however, detection of the spores of *P. larvae* does not allow conclusions to be drawn as to whether the colony is already diseased and an AFB outbreak has occurred [17]. Therefore, it is imperative that laboratory detection of *P. larvae* spores be followed by visual inspection of the colony in question and, in the best-case scenario, resampling to confirm or rule out disease outbreak. If the colony does not show clinical AFB symptoms on visual inspection, such as ropy masses in capped brood cells or black, tightly adhering scales in the lower grooves of the brood cells, an official AFB outbreak cannot be declared in most countries, so the official veterinarian lacks a legal basis for ordering the eradication of the colony. Instead, less harmful methods, such as the shook swarm method for curing colonies of infection (not yet diseased), are advisable and will most likely also be successful [9,12]. The visual inspection of a presumably *P. larvae*-infected colony can be considered a precautionary measure that prevents the negative practical consequences that issue from false-positive laboratory results.

The situation is more difficult with false-negative laboratory results. A colony that is incorrectly diagnosed as free of *P. larvae* spores is usually not retested or visually examined to verify the laboratory diagnosis. Instead, the beekeeper can use the negative laboratory result, for example, to move his colonies to another geographic region or to sell supposedly disease-free colonies. In both cases, the disease spreads, which has large negative impacts on other honey bee colonies and beekeepers. Therefore, it is almost more important to avoid false negatives than false positives in AFB monitoring programs.

Although all three matrices described have been used in surveillance programs for many years [17,18,19,20,21,22,23,26,27,28], a comparative study on their sensitivity and limit of detection of *P. larvae* spores and their suitability in diagnostic practices is still lacking. Here, we report our results on the differential sensitivity of detecting *P. larvae* spores in winter hive debris, seasonal hive debris, adult bees, and brood comb honey obtained using samples spiked with defined numbers of spores. Differences in sensitivity between the different matrices were mainly influenced by the extent to which concomitant bacteria overgrew the culture plates and interfered with the detection of *P. larvae* spores. We identified the most abundant concomitant bacterial species by 16S rRNA gene sequencing and analyzed the possibility of suppressing their growth using culture plates supplemented with antibiotics or by diluting the samples. In addition, we determined the recovery rate, and hence the detection limit, of *P. larvae* spores in samples of brood comb honey or adult bees. Finally, field samples of adult bees and brood comb honey originating from colonies in the vicinity of officially diagnosed AFB outbreaks were tested in parallel to further evaluate the suitability of these two matrices in surveillance programs.

## 2. Materials and Methods

### 2.1. Sample Materials

Samples devoid of *P. larvae* spores were collected from *P. larvae*-free honey bee colonies kept at different locations of the apiary of the Institute for Bee Research (Hohen Neuendorf, Germany). Prior to sampling, all colonies were investigated by thorough visual inspection of every brood comb, and samples of debris, adult bees, and brood comb honey were analyzed via classical microbiology and *P. larvae*-specific PCR analysis [7,14,15]. All analyzed colonies were free of AFB symptoms and *P. larvae* spores and served as sources for collecting *P. larvae*-spore-free hive debris in late summer (seasonal hive debris) or after overwintering (winter hive debris), adult bees, and brood comb honey. To evaluate the suitability of debris, adult bees, and brood comb honey as examination materials for detecting *P. larvae* spores, the *P. larvae*-spore-free samples were spiked with defined numbers of *P. larvae* spores.

In addition, both adult bees and honey were collected from the brood combs of 112 colonies located within the restricted areas of officially diagnosed AFB outbreaks (radius of 3 km around the outbreak site) in North Rhine–Westphalia and Rhineland–Palatinate, Germany, between 2016 and 2019. All samples of adult bees (nurse bees) and hive debris were stored at −20 °C until analysis, while brood comb honey samples were stored at room temperature until analysis.

### 2.2. Bacterial Materials and Spore Preparation

The *P. larvae* type strain ATCC 9545^T^ (genotype ERIC I; [7]) obtained from the American Type Culture Collection (ATCC) was used throughout the study for all experiments involving the addition of *P. larvae* spores. Spore suspensions of ATCC 9545 were prepared essentially as previously described [7]. Briefly, vegetative bacteria were cultivated on Columbia Sheep blood Agar (CSA) plates (Oxoid Deutschland GmbH, Wesel, Germany; containing 5% sterile defibrinated sheep blood) at 37 °C. Colonies of *P. larvae* were resuspended in brain heart infusion (BHI, Merck, Darmstadt, Germany) broth, resulting in a viscous bacterial suspension. BHI broth was added to Columbia sheep blood slant agar tubes, and the viscous bacterial suspension was pipetted to the bottom of the liquid part of the agar slants to create anaerobic conditions. Inoculated slant agar tubes were incubated at 37 °C for at least 10 days to achieve sporulation of *P. larvae*. Subsequently, the lower liquid parts containing the spores were transferred to 1.5 mL tubes and stored at 4 °C until further use. Spore concentrations were determined by cultivating serial dilutions of the harvested spore suspensions on CSA plates. Colony-forming units (cfu) were counted after six days, and spore concentrations were calculated as previously described [29,30].

### 2.3. Detection of P. larvae Spores in Adult Bees or Seasonal/Winter Hive Debris

Published protocols for the detection of *P. larvae* spores in adult bee samples [20,22] or hive debris [31] were adopted in adapted forms and optimized for our laboratory conditions. Briefly, adult bees (25 individuals) or winter/seasonal hive debris (3 g) were transferred to plastic bags (Seward Stomacher bags, Cole-Parmer, Wertheim, Germany) and 12.5 mL autoclaved Milli-Q water (Merck) was added. To evaluate the suitability of adult bees as well as hive debris as sample materials for detecting *P. larvae* spores, *P. larvae*-spore-free samples were spiked with defined numbers of *P. larvae* spores: 125,000 spores, 12,500 spores, 1250 spores, and 125 spores, theoretically resulting in 10,000 cfu/plate, 1000 cfu/plate, 100 cfu/plate, and 10 cfu/plate, respectively. Samples not spiked with *P. larvae* spores were used as controls. Subsequently, the samples were homogenized in a Seward Stomacher 80 Biomaster (Cole-Parmer, Wertheim, Germany) at maximum speed for 6 min. The liquid parts of the samples were filtered (mesh size of 100 µm, Falcon Cell Strainer, Corning, NY, USA) into 50 mL falcon tubes and centrifuged at 18,353× *g* for 10 min. The supernatants were discarded and the pellets resuspended in 2 mL sterile Milli-Q water, resulting in final sample volumes of about 2.5 mL. For seasonal and winter hive debris, all spiked and control samples were also tested in a dilution series (1:10 and 1:100). To inactivate fungal spores [32] and facilitate the germination of *P. larvae* spores [33], undiluted as well as diluted spiked and control samples were heated at 90 °C for 6 min in a water bath. The cooled samples were plated (200 µL/plate) on CSA plates (commercially available from Oxoid Deutschland GmbH) and CNA plates (commercially available from Oxoid Deutschland GmbH; Columbia-CNA agar with 7% sterile defibrinated sheep blood containing 7.5 µg/mL colistin and 5 µg/mL nalidixic acid). Agar plates were incubated at 37° C for six days to allow *P. larvae* spores to germinate and colonies to grow. Thereafter, the plates were examined for the presence (qualitative result) and number (quantitative result) of *P. larvae* colonies. A sample was classified as “*P. larvae*-positive” if at least one *P. larvae* colony on one of the three plates (technical replicates) was detected and identified by PCR [7,14,15]. In addition, the proportion of the plate surface covered by concomitant bacteria of the hive-associated microbiota was estimated and given as a percentage of the entire plate area. All tests were performed with three technical replicates per sample and three independent biological replicates per test. To visualize the results, the values are presented in scatter plots also showing the mean values ± SDs.

To evaluate the effect of increased concentration of nalidixic acid on the growth of concomitant bacteria from debris samples, control samples (seasonal/winter hive debris) not spiked with *P. larvae* spores were plated on CNA plates (Oxoid Deutschland GmbH) and CSA/NA plates prepared with Columbia Sheep blood agar (39 g/L Columbia agar base, 7% sterile defibrinated sheep blood; both obtained from Oxoid Deutschland GmbH) supplemented with 10 µg/mL nalidixic acid (Merck, Darmstadt, Germany). The proportion of the plate surface covered by concomitant bacteria of the hive-associated microbiota was estimated and given as a percentage of the entire plate area. All tests were performed with three technical replicates per sample and three independent biological replicates per test. To visualize the results, the values are presented in scatter plots also showing the mean values ± SDs.

### 2.4. Detection of P. larvae Spores in Brood Comb Honey

For the detection of *P. larvae* spores in brood comb honey, published protocols were used [7,15,34] that correspond to the official protocols of the Office International des Epizooties (OIE) and the German National Reference Laboratory [14,31]. Brood comb honey (3 g) was diluted with 3 mL autoclaved Milli-Q water (Merck). To evaluate the suitability of brood comb honey as a sample material for detecting *P. larvae* spores, defined numbers of *P. larvae* spores were added to the *P. larvae*-spore-free samples: 300,000 spores, 30,000 spores, 3000 spores, and 300 spores, theoretically resulting in 10,000 cfu/plate, 1000 cfu/plate, 100 cfu/plate, and 10 cfu/plate, respectively. Samples not spiked with *P. larvae* spores were used as controls. To inactivate fungal spores [32] and facilitate the germination of *P. larvae* spores [33], spiked and non-spiked samples were heated at 90 °C for 6 min in a water bath. The cooled samples were plated on CSA and CNA plates (200 µL/plate), with three technical replicates per sample. The plates were incubated at 37 °C and examined after six days for the presence (qualitative result) and number (quantitative result for determining the recovery rate) of *P. larvae* colonies. A sample was classified as “*P. larvae*-positive” if at least one *P. larvae* colony on one of the three plates (technical replicates) was detected and identified by PCR [7,14,15]. In addition, the proportion of the plate surface covered by concomitant bacteria of the hive-associated microbiota was estimated and given as a percentage of the entire plate area. All tests were performed in triplicate, i.e., with three biological replicates each. For the samples containing 300,000 and 30,000 spores, the number of *P. larvae* colonies grown on the plates was too high to be counted. To sufficiently lower the number of bacterial colonies per plate, the samples were diluted (1:10, 1:100) prior to plating. To visualize the results, the values are presented in scatter plots also showing the mean values ± SDs.

### 2.5. 16S rRNA Gene Sequencing for Species Identification of Concomitant Bacteria

All sample matrices contained concomitant bacteria from the hive-associated microbiota in varying degrees. To assess the species diversity of the concomitant bacteria in the four matrices (seasonal and winter hive debris, adult bees, and brood comb honey), we collected bacterial colonies from the plates used for the spike experiments. To increase the number and diversity of the samples, we additionally collected brood comb honey from five different honey bee colonies at five different locations of the Institute’s apiary. In total, we were able to draw on about 290 plates for our analyses. Concomitant bacteria growing on the culture plates were documented by pictures taken on the last day of the incubation period of six days. The most abundant, conspicuous, or troublesome (swarming behavior) bacteria were selected for identification via 16S rRNA gene sequencing [35]. To this end, a loop of each selected colony was transferred to a new agar plate and incubated at 37 °C overnight to obtain a pure culture. For DNA extraction, one colony from each pure culture was resuspended in 50 µL double-distilled water and boiled at 95 °C for 15 min. Afterwards, the suspension was centrifuged at 2380× *g* for 5 min and the DNA-containing supernatant was transferred to a new 1.5 mL reaction tube. For species identification, 16S rRNA gene sequencing was performed using the following primers [35]: FD1 (5′-AGAGTTTGATCCTGGCTCAG-3′) and RP2 (5′-ACGGCTACCTTGTTACGACTT-3′). PCR was performed with a final volume of 25 µL containing 1× PCR-buffer (Qiagen, Hilden, Germany), 200 µM of dNTPs (dATP, dCTP, dGTP, and dTTP) (Roth, Karlsruhe, Germany), 0.2 µM of each primer, and 0.5 units of HotStar Taq polymerase (Qiagen). The reaction started with an initial denaturation step at 95 °C for 5 min, followed by 30 cycles of 94 °C for 0.5 min, 56 °C for 0.5 min, and 72 °C for 1.5 min and a final elongation step at 72 °C for 10 min. The success of PCR amplification was checked via agarose gel electrophoresis. PCR purification was performed with the QIAquick PCR Purification Kit (Qiagen), according to the manufacturer’s instructions. The concentrations and purities of the PCR products were checked using a take 3 microvolume plate (Biotek, Winooski, VT, USA) and a Synergy HT microplate reader (Biotek). The amplicons were custom-sequenced by Eurofins Genomics Germany GmbH (Ebersberg, Germany). Sequencing results were analyzed using the Basic local alignment tool (BLAST) [36].

### 2.6. Statistical Analysis

Statistical analysis was performed with Graph Pad Prism 6.07 (San Diego, CA, USA). Data were analyzed for normal distribution with the D’Agostino–Pearson normality test. The Kolmogorov–Smirnov nonparametric test (significantly different when *p* < 0.05) was used to analyze the differences in the plate areas occupied by concomitant bacteria of the hive-associated microbiota between the two different culture conditions (CSA vs. CNA) for all matrices. The unpaired Student’s *t*-test with Welch’s correction (significantly different when *p* < 0.05) was used to analyze the differences in the plate areas occupied by concomitant bacteria depending on the concentrations of nalidixic acid in culture plates. The Kruskal–Wallis nonparametric test with Dunn’s multiple comparisons test (significantly different when *p* < 0.05) were used to analyze (i) the differences between the dilutions of seasonal and winter hive debris in the plate areas occupied by concomitant bacteria and (ii) the differences between adult bee samples and brood comb honey with respect to the recovery rates of *P. larvae* spores. All mean values are given ± standard deviations (SDs).

## 3. Results

### 3.1. Qualitative Detection of P. larvae in Different Matrices Spiked with P. larvae Spores

In the laboratory diagnosis of *P. larvae* as the etiological agent of the notifiable disease AFB, the determination of the presence or absence of *P. larvae*, i.e., a qualitative result, is sufficient, although quantitative data may be useful in some circumstances. To evaluate the suitability of seasonal hive debris, winter hive debris, adult bees, and brood comb honey as matrices for the qualitative detection of *P. larvae*, *P. larvae*-spore-free samples of these matrices were experimentally spiked with varying numbers of *P. larvae* spores which could have yielded between 10 and 10,000 cfu per plate on a purely mathematical basis (Table 1). Non-spiked samples served as controls. Samples were cultivated for six days on CSA and CNA plates. The latter contained 7.5 µg/mL colistin and 5 µg/mL nalidixic acid to suppress the growth of concomitant bacteria and could thus facilitate the germination and growth of *P. larvae*.

Contrary to our expectation of being able to detect *P. larvae* in every sample, *P. larvae* was not detected in any of the debris samples, although samples spiked with the highest spore concentration could have been expected to result in up to 10,000 cfu per plate (Table 1). For both matrices, seasonal and winter hive debris, it made no difference whether the samples had been cultured on CSA or CNA plates; no *P. larvae* colonies were detected in either case, over the entire spore concentration range (Table 1). The situation was more complex for the adult bee samples (Table 1). Cultivation on CSA and CNA plates resulted in the reliable detection of *P. larvae* in all samples containing the highest number of spores, with an expected growth of up to 10,000 cfu per plate. With an expected growth of 1000 cfu per plate, in two of the three biological replicates, all three plates (technical replicates) showed growth of *P. larvae* colonies, while one biological replicate was negative. The spore concentration, which could have resulted in up to 100 cfu per plate, did not give a positive result on CSA plates, but on one CNA plate 11 *P. larvae* colonies were detected, resulting in one of the three biological replicates being classified as “*P. larvae*-positive” (Table 1). The lowest spore load in the adult bee samples, which theoretically could have been as low as 10 cfu per plate, was not detected on either CSA or CNA plates (Table 1). Brood comb honey as a spiked matrix was the only matrix that gave the expected results: all brood comb honey samples, even those with an expected growth of only up to 10 cfu per plate, resulted in the growth, and hence detection, of *P. larvae* colonies (Table 1), and thus gave the correct positive results. None of the control plates had *P. larvae* colonies growing on them (Table 1).

### 3.2. Impact of Concomitant Bacteria in the Spiked Matrices on the Detection of P. larvae

During visual inspection of the plates, we noticed that there were large differences in the growth of concomitant bacteria from the hive-associated microbiota depending on the matrix used (Figure 1A). These differences in concomitant bacteria may differentially impact *P. larvae* detection, simply because of differences in competition with *P. larvae* for space and nutrients. Therefore, we quantified this impact of concomitant bacteria by estimating the proportions of the culture plate surface areas they occupied.

When hive debris as a matrix was cultured on CSA plates, the areas occupied by concomitant bacteria reached nearly 100% for both seasonal hive debris (97.0% ± 7.3%) and winter hive debris (99.4% ± 1.9%), leaving no space for *P. larvae* to germinate and grow (Figure 1B). On CNA plates, the situation for *P. larvae* was not much better, since between 90.7% ± 11.9% (seasonal hive debris) and 96.2% ± 6.8% (winter hive debris) of the culture plate surfaces were covered with concomitant bacteria after six days (Figure 1B). For debris samples, there was no statistically significant difference between cultivation on CSA and CNA plates (Kolmogorov–Smirnov test, *p* > 0.05); hence, culture media supplemented with the antibiotics colistin and nalidixic acid did not solve the problem of concomitant bacteria being present in these matrices and interfering with *P. larvae* detection.

When adult bee samples were used as matrices for *P. larvae* detection, growth of concomitant bacteria was markedly lower and resulted in only 38.6% ± 21.7% (CSA plates) and 20.9% ± 13.7% (CNA plates) of the plate surfaces being covered with interfering bacteria (Figure 1B). The difference in the occupied mean plate surfaces between cultivation on CSA and CNA plates was statistically significant (Kolmogorov–Smirnov test, *p* < 0.001), and, in addition, the number of samples with more than 50% of the plate surface occupied by concomitant bacteria decreased from 14 (on CSA plates) to 2 (on CNA plates) out of 45 (Figure 1B). 

Concomitant bacteria were even less of a problem when samples of brood comb honey were used as matrices, as interfering bacteria normally covered only between 15.3% ± 23.3% (CSA plates) and 4.4% ± 13.1% (CNA plates) of the plate surfaces (Figure 1B). The difference between cultivation on CSA and CNA plates was statistically significant (Kolmogorov–Smirnov test, *p* < 0.01) (Figure 1B). When CSA plates were used, there were five samples for which 50% or more of the plate surface was overgrown by concomitant bacteria, while on the plates of the remaining 40 samples, concomitant bacteria occupied less than 30% of the plate surfaces. On CNA plates, all samples but one yielded culture plates with 20% or less of the surface covered with concomitant bacteria (Figure 1B). Hence, the use of media supplemented with the antibiotics colistin and nalidixic acid positively influenced the detection of *P. larvae* spores in adult bee and brood comb honey samples by reducing the problem posed by concomitant bacteria.

### 3.3. Identification of Concomitant Bacteria with the Different Matrices

In order to assess the nature and diversity of the concomitant bacteria, which complicated correct AFB diagnosis, the most abundant, conspicuous, or troublesome (swarming behavior) bacteria were identified via 16S rRNA gene sequencing (Table 2). With the exception of *Arthrobacter* sp., which is Gram-variable, all identified bacterial species were Gram-positive. Since the laboratory protocols for cultivating *P. larvae* from the different matrices always include a heating step [14,31], it is not surprising that the majority of the identified species were spore formers (Table 2). *Bacillus megaterium*, *B. pumilus*, and members of the *B. cereus* group were the most prevalent species, being present in more than 10% of the samples (Table 2). *B. licheniformis*, *B. velezensis*, *Lysinibacillus fusiformis*, *L. pakistanensis*, *L. sphaericus*, and *Staphylococcus capitis* were present in 5% to 10% of the samples, whereas the others were identified in less than 5% of the samples. Since most of the identified bacterial species are swarmers (Table 2; Figure 2A), it is conceivable that their presence in a sample can cause considerable problems in the laboratory diagnosis of *P. larvae*.

### 3.4. Reducing the Growth of Concomitant Bacteria in Debris Samples with Nalidixic Acid

For adult bee and brood comb honey samples, the differences in the proportions of the plate surfaces covered with concomitant bacteria were statistically significant between cultivation on CSA and CNA plates (Figure 1). However, this was not the case for debris samples (Figure 1). Therefore, in a separate experiment, we attempted to suppress the germination and growth of the troublesome concomitant bacteria in debris by doubling the concentration of nalidixic acid but omitting colistin as a supplement. The use of CSA plates containing 10 µg/mL nalidixic acid (CSA/NA plates) is specified as a culture option for reducing the growth of interfering bacteria in the official collection of methods of the German National Laboratory for Bee Diseases [31]. Growth of concomitant bacteria on commercially obtained CNA plates served as a reference value. Using seasonal hive debris as a matrix resulted in 75.0% ± 7.8% and 82.8% ± 12.5% of the surface areas being occupied by concomitant bacteria on CNA and CSA/NA plates, respectively (Figure 2B). When winter hive debris was used as a matrix, 68.3% ± 15.1% and 57.8% ± 14.0% of the CNA and CSA/NA plate surfaces, respectively, were occupied by concomitant bacteria (Figure 2B). For both seasonal and winter hive debris, there was no statistically significant difference (unpaired Student’s *t*-test with Welch’s correction, *p* > 0.05) in the proportions of the culture plate areas occupied by concomitant bacteria between CNA and CSA/NA plates (Figure 2B), suggesting that further increasing the concentration of nalidixic acid would not solve the diagnostic problem posed by concomitant bacteria.

### 3.5. Reducing the Impact of Concomitant Bacteria in Debris Samples by Sample Dilution

Dilution of a sample is also a way to reduce the impact of concomitant bacteria on *P. larvae* detection. Therefore, we next prepared dilutions (1:10 and 1:100) of the *P. larvae*-spore-spiked seasonal and winter hive debris samples (10 cfu/plate, 100 cfu/plate, 1000 cfu/plate, and 10,000 cfu/plate), as well as of the non-spiked controls, to reduce the germination and growth of concomitant bacteria interfering with *P. larvae* detection (Figure 3). The effect of sample dilution on the concomitant bacteria covering the surface of the culture plates was clearly evident for both matrices on both plate types and for both dilutions, although not all dilutions resulted in statistically significant reductions in the proportions of the plate surfaces covered with concomitant bacteria (Figure 3). For both seasonal and winter hive debris samples, dilution of the sample at a ratio of 1:100 compared to the undiluted sample resulted in a statistically significant decrease (Kruskal–Wallis nonparametric test with Dunn’s multiple comparisons test) in the percentage of culture plate area occupied by concomitant bacteria on both CSA and CNA plates (Figure 3). When undiluted samples were cultured, all plates showed at least 70% (CSA) or 65% (CNA) of the plate area occupied by concomitant bacteria. In contrast, culturing the 1:100 dilutions of the samples on CNA plates resulted in less than 50% of the plate area being occupied by concomitant bacteria in 10 (seasonal hive debris) and 7 (winter hive debris) plates out of 45, respectively, although the mean proportion of the CNA plate area covered with concomitant bacteria was still 72.4% ± 25.9% for seasonal hive debris and 74.8% ± 24.4% for winter hive debris.

We also evaluated all plates for the growth of *P. larvae* colonies. On none of the control plates (non-spiked debris samples; *n* = 108) had *P. larvae* colonies grown. The same was the case for all plates (*n* = 324) on which debris samples were cultured that had been spiked with 12,500, 1250, or 125 *P. larvae* spores.

In seasonal hive debris samples spiked with 125,000 *P. larvae* spores (*n* = 54), *P. larvae* colonies were detected on five plates (Figure 3, green dots) and only in samples that had been diluted 1:100 before streaking on CNA plates. Nevertheless, for this spore concentration and dilution factor, all three biological replicates could have been correctly classified as “*P. larvae*-positive”. Considering the dilution factor, theoretically, 100 *P. larvae* colonies should have grown on these plates, but in fact only one (*n* = 3), three (*n* = 1), and five (*n* = 1) colonies per plate were detected. The proportion of the culture plate area occupied by concomitant bacteria varied between 15% and 75% for the *P. larvae*-positive plates, with four of these five plates having 50% or less of the surface covered with concomitant bacteria (Figure 3).

In winter hive debris samples spiked with 125,000 *P. larvae* spores (*n* = 54), on only one plate (Figure 3, green dot) was one *P. larvae* colony detected, and again this was a sample that had been diluted 1:100 and cultured on a CNA plate with 45% of the culture plate area occupied by concomitant bacteria (Figure 3). These results show that for samples with a very high load of *P. larvae* spores, dilution of the sample at a ratio of 1:100 and culturing on CNA plates can allow the detection of *P. larvae*, although the extremely low number of *P. larvae* colonies suggests that detection depends on chance rather than being robust and reliable. However, even under these conditions (1:100 dilution and presence of nalidixic acid), about half of the culture plates for seasonal hive debris samples (four out of nine) and most of the plates for winter hive debris samples (eight out of nine) gave false-negative results.

### 3.6. Sensitivity of Detection: Recovery Rates in Adult Bee and Brood Comb Honey Samples

In contrast to the results obtained with debris samples, *P. larvae* could be detected in undiluted samples when adult bees or brood comb honey were used as sample matrices spiked with different numbers of *P. larvae* spores (Table 1). In adult bee samples, *P. larvae* was reliably qualitatively detected when the samples had been spiked with 125,000 *P. larvae* spores (Table 1; Figure 4A), although the mean recovery rate was only 3.3% ± 3.1% on CSA and 4.7% ± 4.1% on CNA plates (Figure 4A). It was also possible to detect *P. larvae* when lower spore concentrations were applied (Figure 4A), but this detection was less reliable (Table 1). The detection limit differed between CSA and CNA plates and was 1000 cfu on CSA and 100 cfu on CNA plates, with mean recovery rates of 0.5% ± 1.0% and 1.2% ± 3.5% on CSA and CNA plates, respectively (Figure 4A).

The mean recovery rate of spores in brood comb honey was far better than that in adult bee samples. Even the lowest spore concentration, resulting theoretically in 10 cfu/plate, enabled the detection of *P. larvae*, with mean recovery rates of 101.1% ± 45.1% and 87.8% ± 32.9% on CSA and CNA plates, respectively (Table 1; Figure 4B). The lowest mean recovery rate was 46.8% ± 20.7% for brood comb honey spiked with 300,000 *P. larvae* spores, which could have resulted in 10,000 cfu/plate on CSA plates (Figure 4B). In this group, there was also a plate on which no *P. larvae* colonies had grown (Figure 4B). However, as three aliquots per sample are routinely plated out, the sample would still have been correctly diagnosed as *P. larvae*-positive.

The mean recovery rates of spores present in brood comb honey showed statistically significant differences (Kruskal–Wallis nonparametric testing with Dunn’s multiple comparisons test) between the spore concentrations used for the spiking of the samples (Figure 4B). However, such differences are only relevant if quantification of the spore load is the purpose of the analysis. For the qualitative detection of *P. larvae* spores in routine diagnostics, the most important point is that detection of *P. larvae* was reliably possible on both CSA and CNA plates, even at the lowest concentration tested, i.e., 10 cfu/plate. These results showed that brood comb honey provided the best recovery rates and thus the most sensitive detection of *P. larvae* spores among the matrices tested.

### 3.7. Field Evaluation of Brood Comb Honey and Adult Bees as Examination Materials

The differences between adult bee and brood comb honey samples as matrices in terms of detection limits and recovery rates of *P. larvae* spores were striking, and the question arose whether these would be relevant in the field. Therefore, we compared the use of adult bee and brood comb honey samples in the field diagnosis of *P. larvae* by collecting and analyzing both matrices from the same honey bee colonies (*n* = 112). To increase the chance of obtaining *P. larvae*-positive samples, all sampled colonies were located within the restricted area of officially diagnosed AFB outbreaks (radius of 3 km around the outbreak site).

When brood comb honey was used as a matrix, 67 honey bee colonies tested positive for *P. larvae* spores (Figure 5A) and 45 tested negative (Figure 5B). In contrast, using adult bees as a sample matrix resulted in 76 colonies that tested positive (Figure 5A) and 36 colonies that tested negative (Figure 5B) for *P. larvae* spores. These results suggested that using adult bees as a matrix increased the sensitivity of detecting *P. larvae*-infected colonies, as more colonies were identified as infected than when brood comb honey was used. However, the analysis of the concordance of the *P. larvae*-positive (Figure 5A) and *P. larvae*-negative results (Figure 5B) for the two matrices revealed that it was not simply a matter of more colonies testing positive for *P. larvae* spores with adult bees as a matrix, but rather that, for some colonies, the results differed between the two matrices. Therefore, of the 112 bee colonies sampled, a total of 61 colonies were diagnosed as *P. larvae*-positive and 30 colonies as *P. larvae*-negative using both matrices (Figure 5). When adult bees were used as a matrix, 15 colonies were identified as *P. larvae*-positive that were found to be negative when brood comb honey was analyzed. Similarly, when brood comb honey was used as a matrix, six colonies were identified as *P. larvae*-infected that were diagnosed as “not infected” when adult bees were analyzed (Figure 5).

## 4. Discussion

AFB is a highly contagious and fatal disease in honey bee colonies and a notifiable epizootic in most countries. This disease causes considerable economic losses in apiculture, as diseased but undiagnosed colonies eventually die from AFB, and diagnosed AFB-diseased colonies are often burned, as most authorities consider a rigorous eradication policy the only sustainable control measure [39,40]. In any case, the colonies are lost in one way or another.

However, early diagnosis of infected but not yet diseased colonies opens the possibility of ridding colonies of *P. larvae* spores by the so-called shook swarm or artificial swarm method, in which the brood, the brood combs, and the honey combs, all of which may carry the pathogen, are destroyed and only the adult bees are retained for building a new colony [12]. Artificial swarms are reported to work well with infected colonies but not with diseased colonies [9], underpinning the necessity and importance of identifying infected colonies by appropriate preventive measures before the outbreak of the disease [13]. One such preventive measure is to routinely monitor honey bee colonies for the presence of *P. larvae* spores by analyzing samples, such as debris, adult bees, or brood comb honey, taken from the hive [17]. The success and acceptance of such AFB monitoring programs depends crucially on the correctness of the diagnostic results. Hence, precautions must be taken to avoid both false-positive and false-negative results, and it is almost more important to avoid false negatives in AFB monitoring programs. These considerations provided the rationale for our study, in which we investigated whether different matrices were equally suitable for the detection of *P. larvae* spores and for the avoidance of false-negative results.

In our study, no false-positive results were obtained; however, false-negative results were obtained for all samples spiked with *P. larvae* spores when hive debris was used as a matrix. The false-negative results were independent of the concentration of *P. larvae* spores in the sample as well as the culture medium used. The key factor that apparently hindered the correct detection of *P. larvae* spores was the presence of dominantly growing, concomitant bacteria from the hive microbiota. Germination of *P. larvae* spores on agar plates takes approximately three days, while most concomitant bacteria germinate and their colonies start to grow and occupy plate surfaces within one day. Concomitant bacteria that germinate and grow faster than *P. larvae* appear to restrict nutrients and space to an extent that prevents *P. larvae* germination and growth. The problem of concomitant bacteria was further exacerbated by the fact that most of the identified bacteria were swarmers, so that even a few colonies of these species were able to overgrow the agar plates within a few days. This problem could not be solved by simply using culture plates supplemented with nalidixic acid, the antibiotic recommended by the German National Reference Laboratory for Bee Diseases for reducing the growth of unwanted bacteria in *P. larvae* diagnosis [31]. We first tested commercially available plates (CNA plates) that contain a combination of nalidixic acid and colistin. However, there was no significant effect on the growth of the concomitant bacteria in debris samples when CNA plates were used, indicating that the combination or concentration of antibiotics in the plates used was unsuitable for reducing the growth of the concomitant bacteria. The cyclic cationic peptide antibiotic colistin targets the membranes of Gram-negative bacteria [41] and is thus less or not active against Gram-positive bacteria, which predominated in all hive matrices. Therefore, colistin cannot be expected to significantly affect the growth of concomitant bacteria. With regard to nalidixic acid, it is said that, although Gram-positive bacteria are generally more resistant [42], they can still be inhibited in their growth by this antibiotic [43]. To enhance the effect of nalidixic acid, we also tested self-made culture plates without colistin but with an increased concentration of nalidixic acid (CSA/NA plates), as recommended by the German National Reference Laboratory for Bee Diseases [31], but even this did not significantly reduce the growth of the concomitant bacteria in the debris samples to a point that allowed the detection of *P. larvae*. Only the additional dilution of the samples at 1:100 resulted in a few successful detections of *P. larvae* on five out of nine (seasonal hive debris) and one out of nine (winter hive debris) culture plates, on which sample materials containing the highest concentration of *P. larvae* spores were tested. Hence, the detection of *P. larvae* spores in debris samples depends on high spore concentrations (more than about 40,000 spores per gram of debris) and successful reduction of the growth of concomitant bacteria by combining sample dilution and nalidixic acid-supplemented culture plates. However, even if both conditions are fulfilled, the residual risk of false-negative results, especially when analyzing winter hive debris samples, must not be disregarded. After all, in our study, eight out of nine winter hive debris samples were falsely diagnosed as negative, even though they initially contained approximately 41,000 spores per gram of debris and had been cultured on culture plates with nalidixic acid after a dilution of 1:100. Nevertheless, the use of winter hive debris in particular as a sample matrix may have its justification, as it is often the only material available for AFB diagnosis in winter. However, if brood comb honey or adult bees are available, these matrices should definitely be preferred to debris samples.

Brood comb honey and adult bees are both well-established matrices for the detection of *P. larvae* spores and the diagnosis of not yet clinically obtrusive colonies affected by AFB [18,20,21,22,23,26,44]. While in the case of brood comb honey, the question of sampling location does not really arise, the question of where to sample adult bees is less clear but still relevant, as the probability of contact with *P. larvae* spores differs between forager bees and in-house bees. It has already been shown that bees from the brood comb, honey chamber, and edge frame are better suited for *P. larvae* spore detection than forager bees from the hive entrance [22]. In our field evaluation of brood comb honey and adult bees as examination materials, we collected adult bees from brood frames because these bees are the ones in contact with contaminated brood comb honey and which drive transmission within the colony. In a previous study, comparison of the spore loads of adult bees and honey collected from the brood comb in clinically diseased as well as in infected colonies without clinical signs showed that in both cases adult bees (nurse bees) sampled from the brood comb carried more spores than brood comb honey [21]. This is consistent with our results showing that although detection of *P. larvae* from adult bees in the laboratory experiments was not as sensitive as detection from brood comb honey, samples of adult bees from the brood comb (nurse bees) were equivalent or even superior to brood comb honey samples in the field. However, some false-negative results occurred with both sample matrices, most likely reflecting the differing distribution of the spores in different colonies [19,26]. Therefore, if false negatives must be avoided at all costs, it is best to examine both the adult bees and the honey from the brood combs of a suspect colony. Nevertheless, it should always be kept in mind that false-negative results can never be prevented, and therefore a report must contain an indication that a negative result may also mean that the spore density may simply be below the detection limit.

Although a qualitative result (yes/no) regarding the presence of *P. larvae* spores in a given sample is sufficient to diagnose *P. larvae* infection, there may be issues where a quantitative result is useful. In such cases, brood comb honey should be the preferred matrix, as the recovery rate for *P. larvae* spores was found to be best in this matrix. Even the lowest spore concentration tested was reliably detected, although the standard deviations of the mean values were quite high. However, this problem should be solvable by increasing the number of technical replicates and using dilution series.

In undiluted samples of hive debris plated on CSA or CNA plates, concomitant bacteria tended to cover more than 60% (mean value) of the plate surfaces and detection of *P. larvae* was not possible. In contrast, concomitant bacteria were less of a problem in the detection of *P. larvae* when undiluted samples of adult bees or brood comb honey were cultured on CSA and CNA plates, most probably because concomitant bacteria covered less than 40% (mean value) of the plate surfaces. However, there were also samples in which concomitant bacteria overgrew the plate surface. To reduce the risk of false-negative results in such cases, we recommend classifying a sample as “not suitable for reliable exclusion of *P. larvae* infection” if its cultivation does not yield any *P. larvae* colonies but the surface of the agar plate is predominantly, e.g., more than 50–60%, covered with concomitant bacteria. 

## 5. Conclusions

Brood comb honey was found to be the matrix with the fewest concomitant bacteria and the highest recovery rate and is therefore very suitable as a matrix, especially in AFB surveillance programs that aim at diagnosing colonies infected with *P. larvae* but which are not yet clinically diseased. Adult bee samples were equally well-suited in the field, although in laboratory experiments spore recovery rates were lower than for brood comb honey. The major advantage of adult bees as a sample matrix is that they allow the examination of colonies when no brood is available (e.g., swarms). Based on our results, we cannot recommend the examination of hive debris for the detection of *P. larvae* because the concomitant bacteria dominate samples and make the reliable detection of *P. larvae* much more difficult or even impossible. However, if it is not possible to take samples from the brood comb (e.g., in winter), analysis of winter hive debris may be better than nothing.

## Figures and Tables

**Figure 1 vetsci-10-00103-f001:**
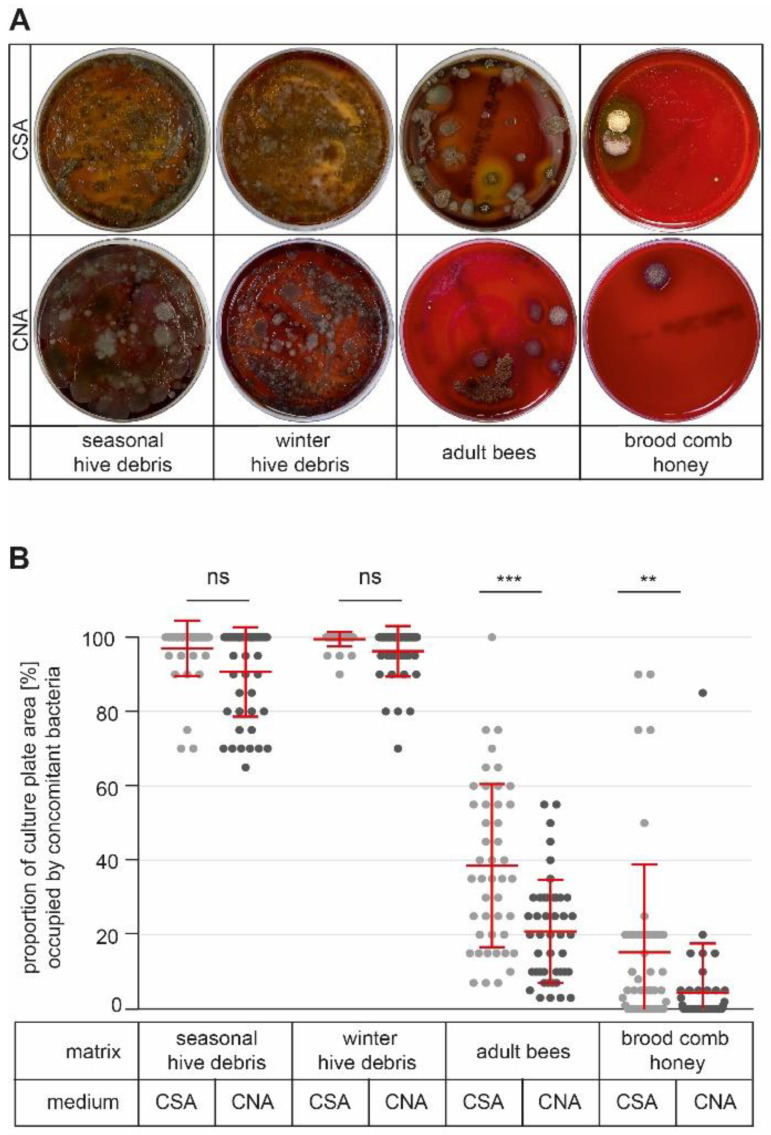
Presence of concomitant bacteria. (**A**) Cultivation of samples from seasonal hive debris, winter hive debris, adult bees, and brood comb honey on CSA and CNA plates always resulted in the growth of concomitant bacteria. Representative pictures from day six of cultivation are shown. (**B**) The proportion of the plate area covered by concomitant bacteria was estimated and expressed as a percentage of the entire plate area. Scatter plots show 45 samples per matrix and culture medium. In addition, mean values ± SDs of the 45 independent samples are given. Data were analyzed by Kolmogorov–Smirnov nonparametric testing (ns, *p* > 0.05; **, *p* < 0.01; ***, *p* < 0.001).

**Figure 2 vetsci-10-00103-f002:**
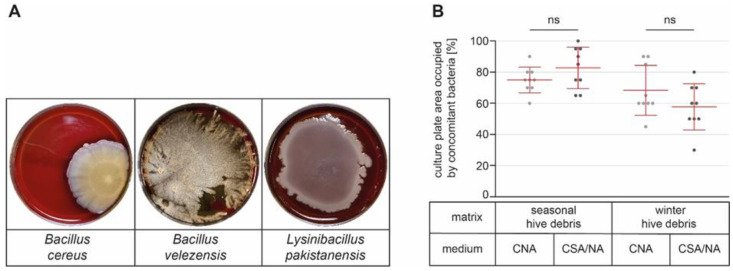
Reducing the growth of concomitant bacteria in debris samples with nalidixic acid. (**A**) Representative pictures of *Bacillus cereus*, *B. velezensis*, and *Lysinibacillus pakistanensis* after six days of culture on CSA plates at 37 °C are shown. (**B**) Samples of seasonal and winter hive debris were each cultivated on CNA plates (CNA) and on CSA plates supplemented with 10 µg/mL nalidixic acid (CSA/NA). The proportion of the culture plate occupied with concomitant bacteria was estimated. Scatter plots show nine samples per matrix and culture medium, as well as the mean values ± SDs of the nine independent samples per group. Data were analyzed by unpaired Student’s *t*-testing with Welch’s correction, and there were no statistically significant (*p* > 0.05) differences between CNA and CSA/NA for both sample types.

**Figure 3 vetsci-10-00103-f003:**
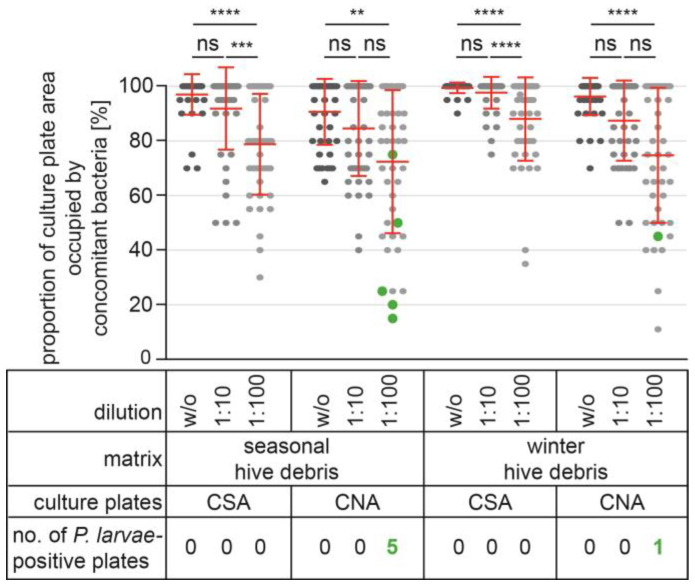
Reducing the impact of concomitant bacteria in debris samples by sample dilution. Samples of seasonal and winter hive debris were spiked with four different amounts of *P. larvae* spores (125,000 spores, 12,500 spores, 1250 spores, and 125 spores; non-spiked samples served as controls) and cultured on CSA and CNA plates either without dilution (w/o) or after being diluted at a ratio of 1:10 or 1:100. Colonies of *P. larvae* were identified, and the proportions of the culture plates occupied with concomitant bacteria were estimated. Scatter plots show 45 samples per undiluted or diluted sample and culture medium, analyzed by Kruskal–Wallis nonparametric testing with Dunn’s multiple comparisons test (ns, *p* > 0.05; **, *p* < 0.01; ***, *p* < 0.001, ****, *p* < 0.0001). In addition, mean values ± SDs of the 45 independent samples are given. The *P. larvae*-positive plates are marked in green.

**Figure 4 vetsci-10-00103-f004:**
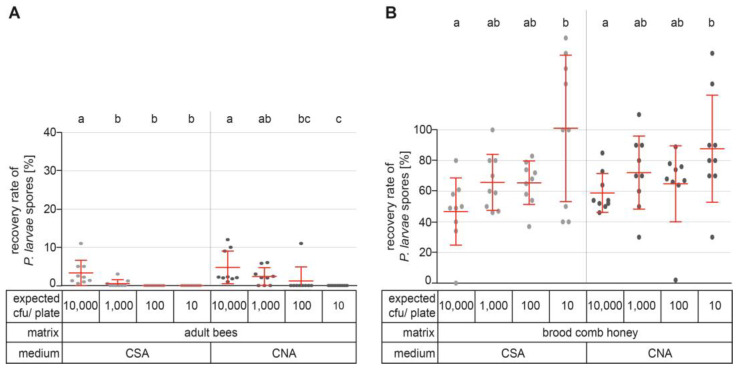
Determination of the recovery rate (%) of *P. larvae* spores in adult bee and brood comb honey samples. (**A**) Adult bee samples and (**B**) brood comb honey samples were spiked with different numbers of *P. larvae* spores, which could have resulted in 10,000 cfu/plate, 1000 cfu/plate, 100 cfu/plate, and 10 cfu/plate. Scatter plots show nine samples each per matrix, culture medium, and expected cfu/plate. Groups with different letters differ significantly, while groups with the same letter are not significantly different (Kruskal–Wallis nonparametric testing with Dunn’s multiple comparison test, with a significance level of *p* < 0.05). In addition, mean values ± SDs of the nine independent samples are given.

**Figure 5 vetsci-10-00103-f005:**
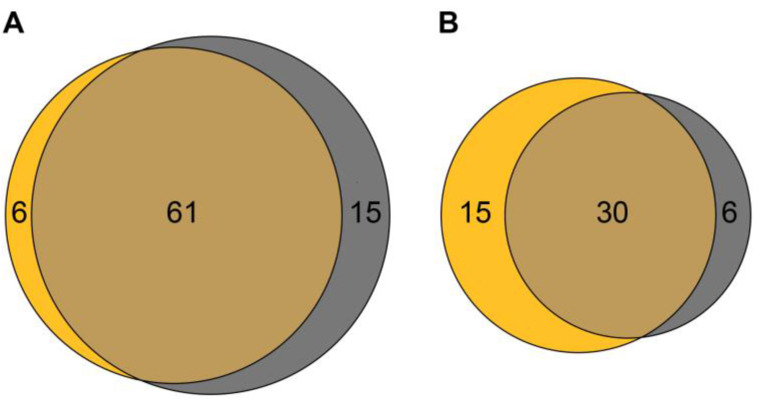
Venn diagram for the comparison of adult bees vs. brood comb honey for use as a matrix in the field. In total, 112 colonies were examined. Brood comb honey samples are visualized in yellow and the corresponding adult bee samples in grey. (**A**) Sixty-one colonies were diagnosed as *P. larvae*-positive using both matrices; 15 colonies and 6 colonies were identified as *P. larvae*-positive only when adult bees and brood comb honey were used as matrixes, respectively. (**B**) Thirty colonies were diagnosed as *P. larvae*-negative using both matrices; 6 colonies and 15 colonies were identified as *P. larvae*-negative only when adult bees and brood comb honey were used as matrixes, respectively. Data were visualized using the online tool: https://eulerr.co/ (accessed on 8 December 2022) [37,38].

**Table 1 vetsci-10-00103-t001:** Detection of *P. larvae* in different matrices spiked with varying numbers of *P. larvae* spores and plated on CSA or CNA culture plates.

Sample Plated on	CSA	CNA	CSA	CNA	CSA	CNA	CSA	CNA	CSA	CNA
Expected result	+ (10,000 cfu/plate)		+ (1000 cfu/plate)		+ (100 cfu/plate)		+ (10 cfu/plate)		Negative control (0 cfu/plate)	
Seasonal hive debris	−	−	−	−	−	−	−	−	−	−
Winter hive debris	−	−	−	−	−	−	−	−	−	−
Adult bees	+	+	(+)	(+)	−	(+)	−	−	−	−
Brood comb honey	+	+	+	+	+	+	+	+	-	-

Note: +, *P. larvae* was detected in all three biological replicates; (+), *P. larvae* was detected in one or two of the three biological replicates; −, *P. larvae* was not detected.

**Table 2 vetsci-10-00103-t002:** Concomitant bacteria identified via 16S rRNA gene sequencing.

Species	Prevalence	Gram Staining	Swarming	Spore-Forming
*Bacillus cereus* group	+++	Pos.	Yes	Yes
*Bacillus megaterium*	+++	Pos.	Yes	Yes
*Bacillus pumilus*	+++	Pos.	Yes	Yes
*Bacillus licheniformis*	++	Pos.	Yes	Yes
*Bacillus velezensis*	++	Pos.	Yes	Yes
*Lysinibacillus fusiformis*	++	Pos.	Yes	Yes
*Lysinibacillus pakistanensis*	++	Pos.	Yes	Yes
*Lysinibacillus sphaericus*	++	Pos.	Yes	Yes
*Staphylococcus capitis*	++	Pos.	No	No
*Arthrobacter* sp.	+	Var.	Yes	?
*Bacillus gibsonii*	+	Pos.	Yes	Yes
*Bacillus mycoides*	+	Pos.	Yes	Yes
*Brevibacillus borstelensis*	+	Pos.	Yes	Yes
*Gordonia polyisoprenivorans*	+	Pos.	No	No
*Lactobacillus kunkeei*	+	Pos.	No	No
*Leifsonia* sp.	+	Pos.	?	No
*Microbacterium oxydans*	+	Pos.	?	No
*Paenibacillus barengoltzii*	+	Pos.	?	Yes
*Paenibacillus glucanolyticus*	+	Pos.	Yes	Yes
*Paenibacillus lautus*	+	Pos.	Yes	Yes
*Paenibacillus odorifer*	+	Pos.	?	Yes
*Paenibacillus polymyxa*	+	Pos.	Yes	Yes
*Rhodococcus* sp.	+	Pos.	No	No
*Sporosarcina* sp.	+	Pos.	?	Yes
*Streptomyces* sp.	+	Pos.	No	Yes

Note: +++, found in >10% of the samples; ++, found in 5–10% of the samples; +, found in <5% of the samples; ?, no conclusive information found in the literature.

## Data Availability

All data generated or analyzed during this study are included in the published article.

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
