# Peer review of "A Comparison of Different Matrices for the Laboratory Diagnosis of the Epizootic American Foulbrood of Honey Bees"

_vetsci, 2023, doi:10.3390/vetsci10020103_

Round 1
Reviewer 1 Report
The authors spiked the P. larvae spores into three standard matrices, including brood comb honey, adult bee, and seasonal/winter hive debris to examine the lowest numbers of spore that can be detected using in vitro propagation system. They suggested that using adult bees and brood comb honey can provide more accurate results, while hive debris showed high false negative results.
Major comment:
It is an interesting research, as the authors confirmed that the contamination of bacteria would inhibit the proliferation of P. larvae spores on culture plate. My concern is why the authors chose using culture plate rather than using PCR as test method, especially the work was performed in the lab. It is understandable that for bee keepers, using plate is a more practical method to examine P. larvae, thus this study can provide some clues for sampling target. However, the plate method showed high false negative result even using P. larvae infected colonies as samples (Line 617-619). I would like to suggest the author provide some explanation why they choose to follow a standard culture procedure, rather than using PCR directly.
Minor comments:
· Line 434, the line numbers from 435 to 467 are overlapped with Table 2, please make sure the Table is clear for reading.
· Line 616, did the authors test the prevalence rate using PCR or other method before the assay? If yes, please describe the method in the materials and methods section. If no, the authors need to explain how they determine the accuracy of their methods and results.
Reviewer 2 Report
This paper explores which sample type is the ideal to diagnose infection with P. larvae in honey bees colonies. Overall, the paper is interesting, well written and statistical analysis are sound. MY major comment is that it would be ideal to combine their culture-dependent results with state-of-the-art diagnosis for P. larvae such as qPCR to measure how much P. larvae growth is inhibited by the concomitant bacteria growth in the culture plate, specially for their field samples. Minor comments are as follows:
Line 45-50: Reference citations are needed for this first paragraph
Lines 53-56. This phrase is a little too long and could benefit of some breaking down for better readability
Lines 54-66: Also missing literature citation here.
Line 89: Consider replacing situation with level or some other word that is more specific.
Line 315: Comma missing after “In the laboratory”.
Lines 314-325: It varies between journals, but I believe that this first paragraph is an element of material and methods and should not be included/repeated here.
Line 354: Avoid non-scientific wording as “huge”
Line 403: Avoid non-scientific wording as “obviously”
Lines 403-407: I believe this would be an element of discussion and not needed here.
Lines 646-651: Citations needed.
Line 658: Delete “already”
Line 676: Here again, avoid the word “obviously”
Line 751: Add OF on “to reduce the risk of false negative”
